# Clarifying the meaning of a positive living environment in nursing homes: A scoping review protocol

Lien Janssens[1], Lisa Geyskens[1,2], Bernadette Dierckx de Casterlé[1], Ellen Vlaeyen[1,3], Mieke Deschodt[1,4]*

1 Department of Public Health and Primary Care, KU Leuven, Leuven, Belgium, 2 Research Foundation–Flanders, Brussels, Belgium, 3 Faculty of Medicine and Life Sciences, Hasselt University, Diepenbeek, Belgium, 4 Competence Center of Nursing, University Hospitals Leuven, Leuven, Belgium

* mieke.deschodt@kuleuven.be

## Abstract

### Background

Nursing homes provide permanent residence and care for older individuals who can no longer live independently. Despite efforts to create a safe and homelike environment, concerns persist regarding residents' well-being. Current research often focuses on different aspects, such as quality of life, sense of home or thriving in the nursing home, each only providing a partial understanding of what constitutes a positive living environment. To address this, we will explore what entails the concept of 'a positive living environment' in nursing homes from the perspectives of residents, staff, and informal caregivers.

### Methods

We will follow Levac and colleagues' scoping review framework, integrating Walker and Avant's concept analysis methodology. Searches will be conducted in PubMed, Embase, Scopus, Web of Science and CINAHL. We will include peer-reviewed qualitative or mixed-method studies, published in English or Dutch in the last 20 years. Studies must address nursing home settings and explore perspectives of residents, staff, or informal caregivers on what constitutes a positive living environment. Title and abstract screening will be performed by one reviewer, with a second reviewer assessing a subset of papers until reaching a 90% agreement rate. Full-text screening will be conducted by one reviewer, with any doubts discussed with the research team. Forward and backward snowballing techniques will be used on papers that pass full-text screening. Data will be extracted and analyzed using concept analysis methodology by examining uses of the concept, defining attributes, antecedents, consequences, and empirical referents. The Preferred Reporting Items

**Data availability statement:** No datasets were generated or analyzed during the current study. All relevant data from this study will be made available upon study completion.

**Funding:** LJ is funded by KU Leuven Internal Funds (C24M/22/047). The funders did not have a role in study design, data collection and analysis, decision to publish, or preparation of the manuscript.

**Competing interests:** The authors have declared that no competing interests exist.

for Systematic Reviews and Meta-Analyses extension for Scoping Reviews will be used as the reporting guideline.

## Conclusion

This protocol outlines the methods for our scoping review to clarify the concept of a positive living environment in nursing homes. Its findings will guide the development of nursing home environments to better meet residents' needs and preferences.

---

### Introduction

The global population is aging rapidly, with the proportion of individuals aged 80 and older expected to double from 4.6% in 2017 to 10.1% by 2050 [1]. As both the total number and proportion of older adults increase, the prevalence of chronic diseases and cognitive impairments is also rising [1]. Consequently, nursing home residents have increasingly complex and demanding care needs. This, in combination with a declining availability of informal caregivers, further intensifies the demand for long-term care facilities, such as nursing homes, to provide permanent residences for older individuals who can no longer live at home [1,2]. Nursing homes often serve as the last home for its residents, underscoring the importance of creating nursing home environments where residents feel safe and at home whilst also receiving adequate care and support [3,4]. In line with this, healthcare organizations are striving to make nursing homes feel more like a home rather than a clinical facility, as seen in the growing emphasis on providing care in small-scale living environments or Green House Homes [5]. These types of facilities move from a strictly clinical care model to a more person-centered care model meaning they are designed to create a homelike feeling while also offering high-quality care [5]. Nonetheless, concerns persist regarding the well-being within these nursing home facilities, as residents are often confronted with challenges such as loneliness, boredom, loss of identity and autonomy and lack of privacy [6–8].

Research addressing these concerns focuses on different concepts related to well-being, such as sense of home, quality of life, quality of care, a positive atmosphere, thriving in the nursing home or the importance of an adapted physical environment [3,8–13]. Such studies provide valuable insights into nursing home residents' preferences and needs within the nursing home setting. They highlight for instance the significance of private space, personal belongings, autonomy, preservation of habits and values, and social interactions in fostering a sense of home in the nursing home [3]. However, while a sense of home seems crucial, it does not encompass the entirety of what constitutes a positive nursing home living environment [4]. Residents also emphasize other aspects such as the importance of receiving good quality care and having staff who possesses the necessary skills to provide and understand their care needs [4].

There has been considerable research focusing on this nursing home living environment, yet studies often focus on different aspects, each providing only a partial

understanding of what constitutes a positive living environment. As a result, there is a variety of terms describing (parts of) the nursing home environment, leaving the concept of a 'positive living environment' vague. Drawing upon existing literature, we aim to identify uses of this concept, the defining attributes, antecedents, consequences, and empirical referents to clarify its meaning [14].

We will conduct a scoping review combined with concept analysis to compile and summarize published information [15]. This approach will guide us to answering our research question: "What entails the concept of 'a positive living environment' in nursing homes from the perspectives of residents, staff, and informal caregivers?". By focusing on the perspectives of those who live in nursing homes, as well as those who care for and interact closely with these nursing home residents, we seek to develop a comprehensive understanding of what a positive living environment entails. This paper outlines the protocol for our scoping review, detailing the methods for searching, selecting, extracting, and synthesizing relevant literature.

## Methods

The scoping review is registered with Open Science Framework (https://osf.io/yc439). The development of this scoping review protocol was informed by the guidance and template provided by Lely and colleagues (2023) [16] as well as the PRISMA-P Checklist (S1 Checklist). The Preferred Reporting Items for Systematic Reviews and Meta-Analyses extension for Scoping Reviews (PRISMA-ScR) will be used as the reporting guideline for the scoping review manuscript [17].

### Design

This review will use the methodological framework described by Lam Wai Shun and colleagues, which combines elements of both the concept analysis and the scoping review methodology [15]. These two methodologies will serve complementary purposes in our review.

The concept analysis methodology of Walker and Avant is a structured approach designed to explore and clarify concepts that are inconsistently used or lack conceptual clarity [14]. It offers structured guidelines for concept clarification through multiple steps: defining uses of the concept, defining attributes, antecedents, consequences, and empirical referents [14]. Nonetheless, this methodology does not specify a strategy for identifying relevant literature.

The scoping review methodology of Levac and colleagues [18] building upon the framework proposed by Arksey and O'Malley [19] provides a structured, transparent, and reproducible approach to systematically map the existing literature on a topic [18]. However, this methodology lacks specific guidelines for analyzing data for concept clarification.

By integrating these two methodologies, we can leverage the strengths of each individual methodology. The scoping review methodology ensures a systematic approach to identify and select relevant publications for our review, while the concept analysis methodology offers a structured framework for clarifying the concept of a 'positive living environment' in nursing homes [15]. Together they align with our goal of comprehensively exploring what constitutes a positive living environment in nursing homes.

### Search strategy

We will use the Peer Review of Electronic Search Strategies (PRESS) as a guide for the development of the search strategy [20]. The search string will incorporate MeSH terms (or equivalent index terms for respective databases) and relevant free text words related to the nursing home setting and our concept 'living environment'. To optimize the search string and ensure its accuracy, we will consult a librarian. Language filters will be applied to include only papers in Dutch and English, with a publication date set from 2004 onwards to cover the last two decades. We will develop the search string tailored for PubMed and subsequently adapt this search string to the other databases.

We will search relevant papers in PubMed (including Medline, via NCBI), Embase (Embase.com), Scopus, CINAHL (via EBSCO) and the following citation indices from the Web of Science Core Collection: Science Citation Index Expanded,

Social Sciences Citation Index, Arts and Humanities Citation Index, Conference Proceedings Citation Index – Science, Conference Proceedings Citation Index – Social Science & Humanities and Emerging Sources Citation index. Furthermore, we will use forward and backward snowballing techniques on papers that pass full-text screening, to identify additional papers. Reference lists of relevant reviews will be scanned as well to identify additional studies.

## Eligibility criteria

We will include peer-reviewed primary research papers published in English or Dutch within the last 20 years that use a qualitative design or a mixed-method design. In case of mixed-method designs, our focus will be specifically on the qualitative component of the study. Literature reviews, conference proceedings and abstracts, editorials, book chapters, protocol papers, quantitative studies, studies evaluating the impact of an intervention, and papers for which the full text is not available will be excluded.

Included studies must focus on a nursing home setting. For the purposes of this scoping review, a nursing home is defined as a facility providing permanent accommodation and care for older adults who can no longer live at home [2]. This care encompasses assistance with household tasks, activities of daily living, and personal care including medical support [2]. Studies focusing on settings where individuals are living (largely) independent and not under constant care (e.g., assisted living accommodations), will be excluded. Additionally, studies examining environments beyond the confines of nursing homes, including external locations such as parks or surrounding areas unrelated to the nursing home premises, will be excluded.

We will include studies that contribute to an understanding of what constitutes a positive living environment for nursing home residents, through the perspective of residents, informal caregivers, and staff, as each provides a unique insight in this concept. Relevant themes include, but are not limited to, the psychological environment (e.g., sense of home), physical environment, social environment, and healthcare services within the nursing home. However, to maintain a focus on broader, non-pandemic-specific factors, studies specifically focusing on the impact of COVID-19 on nursing homes will be excluded. The included studies need to refer to at least one of the following items: uses of the concept, defining attributes, antecedents, consequences, or empirical referents for a positive nursing home living environment [14].

## Study selection and screening

All citations will be imported to EndNote 21.2 Windows (Build: 21.2.0.17387), which will be used for deduplication and citation management. For the study selection and screening, the software tool Rayyan will be used [21]. One reviewer will assess the titles and abstracts of papers based on the predefined eligibility criteria. A second reviewer will assess the first 200 papers with the goal of a minimum agreement rate of 90% [22]. If the minimum agreement is not reached, eligibility criteria will be reevaluated, and another 200 papers will be assessed until 90% agreement is reached. During title and abstract screening, papers that do not contain an abstract (title only) will automatically pass on to full-text screening unless the title indicates the content is unrelated to our topic, in which case it will be excluded.

Subsequently, one reviewer will screen the full text of the included papers against the eligibility criteria to select the final papers for inclusion. When there is doubt whether to include a paper, this will be discussed with the research team. The selection process will be iterative, with any potential adjustments to the eligibility criteria or the selection process documented in the final manuscript.

## Data extraction

A customized data extraction form will be developed to extract all relevant data from each included study. This form will include specific details about the study and study characteristics: first author, publication year, country, aims, research questions relevant to the scoping review, study design, setting, population and sample characteristics, and in- and exclusion criteria. Additionally, the steps congruent with Walker and Avant's concept analysis approach [14] will be included

to extract data related to our concept 'positive living environment': uses of the concept, defining attributes, antecedents, consequences, and empirical referents [14]. Incomplete or missing information will be coded as 'not reported' and confusing information as 'unclear'.

To facilitate this process, we will first create one-page conceptual sheets for each individual study. In these sheets we will extract the most relevant information while considering the specific context of each study. Through this analysis, we will identify relevant elements and classify them as defining attributes, antecedents, consequences or empirical referents. Consequently, the data extraction and synthesis phases will be iterative.

The data extraction form will be tested using five included full-text papers. Given the iterative nature of data extraction and synthesis, the form will be refined, updated, or clarified as needed until all authors reach a consensus on the final version. One reviewer will continue data extraction, with a second reviewer validating the results. Any discrepancies will be addressed through discussion with the research team.

## Sensitivity analysis

Before synthesizing the results, we will conduct a sensitivity analysis to evaluate the relative contribution of each study to the scoping review, considering both its relevance and methodological quality [23]. Relevance will be assessed by the research team based on the study's contribution to the central research question and will be scored as low, moderate, or highly relevant. Methodological quality will be assessed using the JBI Critical Appraisal Checklist for Qualitative Research [24], scoring each study as low, moderate, or high quality. Studies with high contribution will be prioritized in the thematic analysis.

## Synthesis and presentation of results

The data synthesis process will include a descriptive numerical and qualitative thematic analysis [15,18]. Levac and colleagues' scoping review methodology will be used for the numerical analyses, describing study characteristics such as the total number of included studies, types of study designs, (trends in) years of publication, geographical distribution, and study populations [18]. The concept analysis methodology of Walker and Avant will be used to assess the current understanding of the concept [14]. Using Walker and Avants' step-by-step approach, we will explore how the concept of 'living environment' is defined and used in research (uses of the concept), particularly focusing on identifying essential characteristics or features of a positive living environment within nursing homes (defining attributes). Furthermore, this method enables us to pinpoint the essential conditions for fostering a positive living environment (antecedents) and to examine the outcomes associated with its presence (consequences). Finally, we will identify observable indicators that provide evidence of the existence a positive living environment (empirical referents) [14]. The foundation for this analysis will be the one-page conceptual sheets developed for each included study (see 'Data extraction' section). These conceptual sheets will be used to conduct a cross-case analysis guided by the Qualitative Analysis Guide of Leuven (QUAGOL) [25]. Through groups discussions with the research team, we will refine these conceptual sheets based on emerging insights, to ensure that all relevant information is captured.

This approach will help us systematically differentiate between the various elements of the concept analysis while also distinguishing between core aspects and context-specific elements, ensuring that cultural influences are embedded in the analysis rather than treated as separate factors. This approach will help to refine, evaluate and achieve consensus regarding the findings.

The process of identification, selection and exclusion of papers will be visualized in a PRISMA flow diagram [26]. The results of the numerical analysis will be presented in a table format with additional textual explanations. If the results of the concept analysis permit, we will develop a conceptual model to visually represent the results. Additionally, the findings of the concept analysis will be described in detail for each step of the analysis to provide a comprehensive summary. It is anticipated that our approach to presenting the results may evolve as we progress through the review process.

## Discussion

This scoping review protocol outlines our approach to clarify the concept of a positive living environment in nursing homes. We will systematically search for relevant literature following Levac and colleagues' scoping review methodology [18], using multiple databases covering medical and healthcare perspectives (i.e., PubMed, Embase, CINAHL) to broader interdisciplinary views (i.e., Scopus, Web of Science). With a focus on selecting qualitative research, the scoping review will aim to provide a comprehensive overview of stakeholder experiences and perspectives, primarily from residents, supplemented by insights from nursing home staff and informal caregivers. Integrating these diverse viewpoints will help capture the complex and subjective nature of what constitutes a positive living environment in nursing homes.

We will integrate Walker and Avant's concept analysis methodology with Levac and colleague's scoping review methodology [14,15], as the latter lacks clear guidelines for data extraction and qualitative analysis. This integration allows us to thoroughly explore and clarify our concept of a positive living environment, which in turn will inform policy, researchers and healthcare workers to develop targeted interventions and quality improvement projects in nursing homes. More specifically, the defining attributes will indicate which organizational, team or process related components can be improved to create or maintain a positive living environment, whereas the antecedents will provide guidance on context-specific implementation strategies tailored to the needs of staff and residents, such as the necessary training programs for these attributes to manifest. Furthermore, the identification of the consequences will help set measurable goals and objectives for future improvements, while the empirical referents can enable the development of tools and methods to evaluate and monitor the quality of the living environment in nursing homes, enabling continuous progress tracking and necessary adjustments. So, by using the combined methodology of scoping review and concept analysis we can establishes a clear framework for future studies as well as for policy decisions. This ensures that efforts to foster positive living environments are based on well-defined, evidence-based and measurable criteria.

Nonetheless, this scoping review will have some limitations. The scoping review will solely focus on literature published within the last 20 years. While this excludes historical perspectives, the review aims to provide insights into the most relevant and current practices and policies in nursing home settings. Significant demographic shifts such as rapid population aging, the rising prevalence of chronic diseases, and the decreasing availability of family caregivers due to higher workforce participation among women and declining birth rates have reshaped the landscape of the nursing home [1]. Recent literature more accurately reflects these evolving challenges and needs. By focusing on the past two decades, the review ensures that the findings are aligned with current trends and offer insights into the development of effective interventions and tools that address both present and emerging issues within the nursing home. Related to this, we will exclude studies specifically focusing on the COVID-19 pandemic. While the pandemic introduced unique challenges, our goal is to identify core elements of a positive living environment under typical conditions. Pandemic-related issues may highlight temporary or atypical aspects of nursing home life, limiting generalizability. Yet future research could explore the impact of pandemics and similar crises on the nursing home living environment.

Further, based on the language proficiencies of the research team members, we will only include Dutch and English papers. This might introduce cultural bias meaning that our findings may not fully represent perspectives from all cultural contexts. To address this, we will report the country of each included study, providing insights and transparency regarding the cultural scope of our findings. Additionally, the iterative process of data extraction and analysis, using the conceptual sheets and conducting cross-case analysis, ensures that our findings will reflect both generalizable elements and context-specific nuances. This allows us to identify the shared core themes while acknowledging cultural differences. By maintaining flexible and open to emerging insights throughout the analysis, we aim to enhance the rigor and comprehensiveness of our findings. Furthermore, the involvement of multiple reviewers with diverse backgrounds, including nursing, geriatrics, psychology and qualitative research expertise, ensures a broad range of perspectives and minimizes individual bias and interpretation bias as much as possible.

Moving forward, the findings of the scoping review will be disseminated through a peer-reviewed publication, as well as presentations at scientific conferences and network events. Furthermore, the scoping review is part of a broader research project in Flanders, where additional qualitative studies, including interviews, focus groups, and workshops with various stakeholders, will explore the factors that contribute to a positive living environment in Flemish nursing homes. These future studies will help contextualize the findings within the Flemish setting, while validating and expanding upon the insights gained from this scoping review. By broadly sharing our results, we aim to contribute to ongoing efforts aimed at improving nursing home residents' well-being.

## Supporting information

**S1 Checklist.  PRISMA-P (Preferred Reporting Items for Systematic review and Meta-Analysis Protocols) 2015 checklist: recommended items to address in a systematic review protocol.**
(DOC)

## Author contributions

**Conceptualization:** Lien Janssens, Bernadette Dierckx de Casterlé, Ellen Vlaeyen, Mieke Deschodt.

**Funding acquisition:** Mieke Deschodt.

**Methodology:** Lien Janssens, Bernadette Dierckx de Casterlé, Ellen Vlaeyen, Mieke Deschodt.

**Supervision:** Mieke Deschodt.

**Writing – original draft:** Lien Janssens.

**Writing – review & editing:** Lien Janssens, Lisa Geyskens, Bernadette Dierckx de Casterlé, Ellen Vlaeyen, Mieke Deschodt.

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
