## [Decision Letter · Decision Letter 0]

28 Jan 2025

PONE-D-24-34371Clarifying the meaning of a positive living environment in nursing homes: a scoping review protocolPLOS ONE

Dear Dr. Deschodt,

Thank you for submitting your manuscript to PLOS ONE. After careful consideration, we feel that it has merit but does not fully meet PLOS ONE’s publication criteria as it currently stands. Therefore, we invite you to submit a revised version of the manuscript that addresses the points raised during the review process.

The manuscript is well-structured and provides a strong basis for the scoping review, but a few revisions are recommended to enhance its rigor and comprehensiveness:

1. Justification of Study Exclusions: Strengthen the rationale for excluding studies older than 20 years and non-English/Dutch papers, as these exclusions could restrict the reviews comprehensiveness and applicability.

2. Exclusion of Pandemic-Specific Studies: Provide a clearer explanation for excluding pandemic-specific studies, given their potential relevance in understanding positive living environments under stress.

3. Bias Considerations: Expand the discussion on potential interpretation and selection biases, and describe strategies to mitigate them.

4. Validation with Stakeholders: Include a plan to validate findings through feedback from nursing home residents, staff, or informal caregivers to ensure practical relevance and applicability.

5. Practical Applications: Clearly outline how the findings will contribute to the development of practical interventions, policies, or frameworks, highlighting their potential impact on real-world settings.

These revisions will strengthen the manuscripts methodological transparency, practical relevance, and contribution to the field.

We look forward to receiving your revised manuscript.

Kind regards,

Muhammad Shahzad Aslam, Ph.D.,M.Phil., Pharm-D

Academic Editor

PLOS ONE

Journal Requirements:

Reviewers' comments:

Reviewer's Responses to Questions

**Comments to the Author**

1. Does the manuscript provide a valid rationale for the proposed study, with clearly identified and justified research questions?

Reviewer #1: Yes

Reviewer #2: Yes

Reviewer #3: Yes

2. Is the protocol technically sound and planned in a manner that will lead to a meaningful outcome and allow testing the stated hypotheses?

Reviewer #1: Yes

Reviewer #2: Yes

Reviewer #3: No

3. Is the methodology feasible and described in sufficient detail to allow the work to be replicable?

Reviewer #1: Yes

Reviewer #2: Yes

Reviewer #3: Yes

4. Have the authors described where all data underlying the findings will be made available when the study is complete?

Reviewer #1: Yes

Reviewer #2: Yes

Reviewer #3: Yes

5. Is the manuscript presented in an intelligible fashion and written in standard English?

Reviewer #1: Yes

Reviewer #2: Yes

Reviewer #3: Yes

6. Review Comments to the Author

You may also provide optional suggestions and comments to authors that they might find helpful in planning their study.

Reviewer #1: The protocol demonstrates several strengths that enhance its suitability for publication. Notably, it effectively integrates scoping review and concept analysis methodologies, ensuring a comprehensive and nuanced exploration of the concept of a positive living environment in nursing homes. The inclusion of perspectives from multiple stakeholders, such as residents, staff, and informal caregivers, strengthens the depth and breadth of the analysis. The use of well-established frameworks, such as Levac et al.’s scoping review methodology and Walker and Avant’s concept analysis approach, provides a robust methodological foundation. Additionally, the registration of the study on the OSF and the inclusion of the PRISMA-ScR checklist reflect adherence to transparency and reporting standards.

However, there are a few areas where the protocol could benefit from further refinement. While the decision to include only studies published in English and Dutch is acknowledged, a more explicit justification for this limitation would help address potential selection bias. The authors should also elaborate on strategies to mitigate interpretation bias, such as triangulation or external validation methods. Given the cultural variability in perceptions of a positive living environment, a discussion of how the findings might generalize to diverse cultural contexts would further enhance the protocol’s applicability. Lastly, while dissemination plans are mentioned, providing more detail on how the findings will influence policy or practice in nursing home settings would strengthen the discussion and highlight the study’s potential impact.

These adjustments will help refine the protocol and ensure its relevance and rigor in contributing to the field of nursing home care.

Reviewer #2: The manuscript is well-prepared and establishes a strong foundation for the scoping review. However, minor revisions are needed to justify its methodological choices and address potential biases. Here's my suggestion:

1. The exclusion of studies older than 20 years and non-English/Dutch papers may limit the comprehensiveness. Justify these exclusions more robustly.

2. Clarify the rationale for excluding pandemic-specific studies, as they provide relevant insights into positive living environments under stress.

3. Address interpretation and selection biases more thoroughly in the discussion.

4. To ensure practical relevance, include a plan for validating findings with nursing home residents, staff, or informal caregivers.

5. Explicitly outline how findings will inform the development of practical interventions or frameworks.

Reviewer #3: Abstract

Please make the background concise, and present the the gap in the literature by presenting other relevant concepts like safe environment, safe space.

Introduction:

Present the importance of nursing home. Present relevant epidemiologic/statistical data with regards to aging population, patients of nursing homes, and nursing homes.

Present existing relevant concepts of nursing homes.

Methodology:

Looking into the purpose and overall aim of the study, I believe that a scoping review is the least appropriate method to use compared to methods like concept analysis of Walker and Avant and integrative review by Whittemore and Knafl or evolutionary methods.

Please review the different types of review using these suggested links:

Systematic review and scoping review:

https://link.springer.com/content/pdf/10.1186/s12874-018-0611-x.pdf

Concept analysis:

https://www.researchgate.net/profile/Judith-Hupcey/publication/14198489_Concept_analysis_in_nursing_research_A_critical_appraisal/links/5654b6c808aeafc2aabc0093/Concept-Analysis-in-Nursing-Research-A-Critical-Appraisal.pdf

https://www.researchgate.net/profile/Lisbeth-Fagerstroem/publication/49620753_Rodgers%27_evolutionary_concept_analysis-A_valid_method_for_developing_knowledge_in_nursing_science/links/5a8e79d0a6fdcc808c0f8d22/Rodgers-evolutionary-concept-analysis-A-valid-method-for-developing-knowledge-in-nursing-science.pdf

Integrative review:

https://www.researchgate.net/profile/Lisbeth-Fagerstroem/publication/49620753_Rodgers%27_evolutionary_concept_analysis-A_valid_method_for_developing_knowledge_in_nursing_science/links/5a8e79d0a6fdcc808c0f8d22/Rodgers-evolutionary-concept-analysis-A-valid-method-for-developing-knowledge-in-nursing-science.pdf

7. PLOS authors have the option to publish the peer review history of their article (what does this mean? ). If published, this will include your full peer review and any attached files.

**Do you want your identity to be public for this peer review?** For information about this choice, including consent withdrawal, please see our Privacy Policy .

Reviewer #1: **Yes: ** Noriel Calaguas

Reviewer #2: No

Reviewer #3: No

---

## [Author Response · Author response to Decision Letter 1]

28 Feb 2025

Reviewer #1:

The protocol demonstrates several strengths that enhance its suitability for publication. Notably, it effectively integrates scoping review and concept analysis methodologies, ensuring a comprehensive and nuanced exploration of the concept of a positive living environment in nursing homes. The inclusion of perspectives from multiple stakeholders, such as residents, staff, and informal caregivers, strengthens the depth and breadth of the analysis. The use of well-established frameworks, such as Levac et al.’s scoping review methodology and Walker and Avant’s concept analysis approach, provides a robust methodological foundation. Additionally, the registration of the study on the OSF and the inclusion of the PRISMA-ScR checklist reflect adherence to transparency and reporting standards.

However, there are a few areas where the protocol could benefit from further refinement. While the decision to include only studies published in English and Dutch is acknowledged, a more explicit justification for this limitation would help address potential selection bias. The authors should also elaborate on strategies to mitigate interpretation bias, such as triangulation or external validation methods. Given the cultural variability in perceptions of a positive living environment, a discussion of how the findings might generalize to diverse cultural contexts would further enhance the protocol’s applicability. Lastly, while dissemination plans are mentioned, providing more detail on how the findings will influence policy or practice in nursing home settings would strengthen the discussion and highlight the study’s potential impact.

These adjustments will help refine the protocol and ensure its relevance and rigor in contributing to the field of nursing home care

Dear reviewer, thank you for your valuable suggestions.

Regarding the suggestion about the language restrictions, this decision was primarily based on the language proficiencies of the research team. We acknowledge that the focus on English and Dutch literature can introduce a selection/cultural bias. We will include an overview of the countries where the primary research results stem from in the data analysis of the scoping review, meaning we will be transparent about the cultural contexts of the included studies. Further, our findings will account for the cultural variability in perceptions of a positive living environment, by using an iterative approach during data extraction and analysis (see additional section in manuscript, included below). This allows us to distinguish between core elements and context-specific elements, ensuring that cultural influences are embedded in the analysis rather than treated as separate factors.

To mitigate interpretation bias, we will hold multiple group discussions within the research team (peer review). The research team consists of members with diverse backgrounds including nursing, psychology, and expertise in qualitative research. This in combination with the iterative process of data extraction and analysis, ensures that no relevant information is overlooked. By integrating multiple perspectives and staying close to the original data — which itself reflects the views of staff, residents, and informal caregivers—, we aim to minimize interpretation bias while maintaining flexible and open to emerging insights.

We clarified these elements in the methodology and added justification for these limitations in the following sections:

Data extraction section (page 7, lines 162-170)

‘A customized data extraction form will be developed to extract all relevant data from each included study. …

To facilitate this process, we will first create one-page conceptual sheets for each individual study. In these sheets we will extract the most relevant information while considering the specific context of each study. Through this analysis, we will identify relevant elements and classify them as defining attributes, antecedents, consequences or empirical referents. Consequently, the data extraction and synthesis phases will be iterative.

The data extraction form will be tested using five included full-text papers. Given the iterative nature of data extraction and synthesis, the form will be refined, updated, or clarified as needed until all authors reach a consensus on the final version.’

Synthesis and presentation of results section (page 9, lines 192-200):

‘The foundation for this analysis will be the one-page conceptual sheets developed for each included study (see ‘Data extraction’ section). These conceptual sheets will be used to conduct a cross-case analysis guided by the Qualitative Analysis Guide of Leuven (QUAGOL) [25]. Through groups discussions with the research team, we will refine these conceptual sheets based on emerging insights, to ensure that all relevant information is captured. This approach will help us systematically differentiate between the various elements of the concept analysis while also distinguishing between core aspects and context-specific elements, ensuring that cultural influences are embedded in the analysis rather than treated as separate factors. This approach will help to refine, evaluate and achieve consensus regarding the findings.’

Discussion section (page 11, lines 247-258):

‘Further, based on the language proficiencies of the research team members, we will only include Dutch and English papers. This might introduce cultural bias, meaning that our findings may not fully represent perspectives from all cultural contexts. To address this, we will report the country of each included study, providing insights and transparency regarding the cultural scope of our findings. Additionally, the iterative process of data extraction and analysis, using the conceptual sheets and conducting cross-case analysis, ensures that our findings will reflect both generalizable elements and context-specific nuances. This allows us to identify the shared core themes while acknowledging cultural differences. By maintaining flexible and open to emerging insights throughout the analysis, we aim to enhance the rigor and comprehensiveness of our findings. Furthermore, the involvement of multiple reviewers with diverse backgrounds, including nursing, geriatrics, psychology and qualitative research expertise, ensures a broad range of perspectives and minimizes individual bias and interpretation bias as much as possible.’

Regarding the impact of our findings on policy and practice, we outlined in the discussion section how each component of the concept analysis can inform policy development and practical implementation. We have expanded this section to further emphasize the connection between our findings and their applicability to nursing home policies and care strategies.

We adapted the discussion section accordingly (page 10, lines 217-232) :

‘We will integrate Walker and Avant’s concept analysis methodology with Levac and colleague’s scoping review methodology [14, 15], as the latter lacks clear guidelines for data extraction and qualitative analysis. This integration allows us to thoroughly explore and clarify our concept of a positive living environment, which in turn will inform policy, researchers and healthcare workers to develop targeted interventions and quality improvement projects in nursing homes. More specifically, the defining attributes will indicate which organizational, team or process related components can be improved to create or maintain a positive living environment, whereas the antecedents will provide guidance on context-specific implementation strategies tailored to the needs of staff and residents, such as the necessary training programs for these attributes to manifest. Furthermore, the identification of the consequences will help set measurable goals and objectives for future improvements, while the empirical referents can enable the development of tools and methods to evaluate and monitor the quality of the living environment in nursing homes, enabling continuous progress tracking and necessary adjustments. So by using the combined methodology of scoping review and concept analysis we can establishes a clear framework for future studies as well as for policy decisions. This ensures that efforts to foster positive living environments are based on well-defined, evidence-based and measurable criteria.’

We hope these adjustments strengthen the manuscript and address your concerns.

Reviewer #2:

The manuscript is well-prepared and establishes a strong foundation for the scoping review. However, minor revisions are needed to justify its methodological choices and address potential biases. Here's my suggestion:

Dear reviewer, thank you for your valuable suggestions. Please find below our answers and adjustments we have carried out based on your feedback.

1. The exclusion of studies older than 20 years and non-English/Dutch papers may limit the comprehensiveness. Justify these exclusions more robustly.

We elaborated on the exclusion criteria of studies older than 20 years in the discussion section (page 10, lines 233-242):

‘Nonetheless, this scoping review will have some limitations. The scoping review will solely focus on literature published within the last 20 years. While this excludes historical perspectives, the review aims to provide insights into the most relevant and current practices and policies in nursing home settings. Significant demographic shifts such as rapid population aging, the rising prevalence of chronic diseases, and the decreasing availability of family caregivers due to higher workforce participation among women and declining birth rates have reshaped the landscape of the nursing home [1]. Recent literature more accurately reflects these evolving challenges and needs. By focusing on the past two decades, the review ensures that the findings are aligned with current trends and offer insights into the development of effective interventions and tools that address both present and emerging issues within the nursing home.’

The rationale for including only English and Dutch studies is addressed above, in the first paragraph of our response to reviewer 1 (see also page 11, lines 247-258 in the manuscript).

2. Clarify the rationale for excluding pandemic-specific studies, as they provide relevant insights into positive living environments under stress.

The decision to exclude studies specifically focusing on the impact of the COVID-19 pandemic on nursing homes was made to maintain a focus on the broader, non-pandemic-specific factors that contribute to a positive living environment. While the pandemic has undoubtedly introduced unique challenges, our aim is to identify the core elements that define a positive living environment under typical conditions. Studies focusing on pandemic-specific issues such as lockdowns, visitor restrictions, and infection control measures may highlight temporary or atypical aspects of nursing home life that are not representative of typical circumstances. By excluding these studies, we seek to gain a clearer and more robust understanding of the fundamental attributes, antecedents, consequences, and empirical referents of a positive living environment. This approach allows us to provide insights that are generally applicable, rather than specific to crisis situations. We acknowledge that this exclusion is a limitation and that future research could explore the impact of pandemics and similar crises on the concept of a positive living environment, building on the findings of this review. However, within the scope of this study, we focus on defining what constitutes a positive living environment in a general sense. During data analysis, we will consider the pandemic's impact by examining potential shifts in the literature published before and after COVID-19.

We added this justification in the discussion section (page 11, lines 242-246):

‘Related to this, we will exclude studies specifically focusing on the COVID-19 pandemic. While the pandemic introduced unique challenges, our goal is to identify core elements of a positive living environment under typical conditions. Pandemic-related issues may highlight temporary or atypical aspects of nursing home life, limiting generalizability. Yet future research could explore the impact of pandemics and similar crises on the nursing home living environment.’

3. Address interpretation and selection biases more thoroughly in the discussion.

For interpretation and selection biases, we have outlined our strategies in the response above, in the first and second paragraph of our response to reviewer 1. Further details are provided in the manuscript’s discussion section (page 11, lines 247-258)

4. To ensure practical relevance, include a plan for validating findings with nursing home residents, staff, or informal caregivers.

The findings of the scoping review will mainly be validated with a research team, consisting of members with backgrounds in nursing, relevant expertise in research, and hands-on experience in the nursing home context. Importantly, the primary research included in this review is based on qualitative data collected directly from these stakeholder groups (residents, staff and informal caregivers), meaning that our findings are already grounded in their perspectives. For this specific study, we do not plan to formally validate the results with additional residents, staff, or informal caregivers. However, the scoping review is part of a broader research project in Flanders, were we will conduct qualitative studies, including interviews, focus groups and workshops with various stakeholders, to explore what factors contribute to a positive living environment in Flemish nursing homes for residents.

We now mention in the protocol that the scoping review is part of a broader research project (manuscript page 11, lines 260-266)

‘Moving forward, the findings of the scoping review will be disseminated through a peer-reviewed publication, as well as presentations at scientific conferences and network events. Furthermore, the scoping review is part of a broader research project in Flanders, where additional qualitative studies, including interviews, focus groups, and workshops with various stakeholders, will explore the factors that contribute to a positive living environment in Flemish nursing homes. These future studies will help contextualize the findings within the Flemish setting, while validating and expanding upon the insights gained from this scoping review. By broadly sharing our results, we aim to contribute to ongoing efforts aimed at improving nursing home residents’ well-being.’

5. Explicitly outline how findings will inform the development of practical interventions or frameworks.

Regarding the impact of our findings on policy and practice we refer, to the final paragraph of our response to reviewer 1. We added additional justification based on this suggestion within the discussion section (page 10, lines 217-232)

We hope these adjustments strengthen the manuscript and address your concerns.

Reviewer #3:

Abstract:

Please make the background concise, and present the gap in the literature by presenting other relevant concepts like safe environment, safe space.

We added relevant concepts in relation to a positive living environment in background part of the abstract:

‘Background: Nursing homes provide permanent residence and care for older individuals who can no longer live independently. Despite efforts to create a safe and homelike environment, concerns persist regarding residents’ well-being. Current research often focuses on different aspects, such as quality of life, sense of home or thriving in the nursing home, each only providing a partial understanding of what constitutes a positive living environment. To address this, we will explore what entails the concept of ‘a positive living environment' in nursing homes from the perspectives of residents, staff, and informal caregivers.’

Introduction:

Present the importance of nursing home. Present relevant epidemiologic/statistical data with regards to aging population, patients of nursing homes, and nursing homes.

We have incorporated relevant epidemiologic data on the aging population and nursing home residents, including the increa

---

## [Decision Letter · Decision Letter 1]

8 Apr 2025

PONE-D-24-34371R1Clarifying the meaning of a positive living environment in nursing homes: a scoping review protocolPLOS ONE

Dear Dr. Deschodt,

Thank you for submitting your manuscript to PLOS ONE. After careful consideration, we feel that it has merit but does not fully meet PLOS ONE’s publication criteria as it currently stands. Therefore, we invite you to submit a revised version of the manuscript that addresses the points raised during the review process.

The integration of Walker and Avant’s concept analysis with the scoping review methodology is a thoughtful and well-structured approach. The protocol clearly outlines the rationale for the inclusion and exclusion criteria, and the planned strategies to mitigate bias are commendable. Additionally, the emphasis on how findings will inform future research, policy, and practice—particularly through follow-up work in Flanders—further strengthens the relevance and utility of this study.

Pease include a brief description of the **concept analysis methodology** and present it clearly before introducing the **scoping review** framework. Clarify the relationship between the two methods and explain how they complement each other in the context of your research objectives. This addition will enhance the clarity and coherence of the protocol, especially for readers less familiar with either approach.

We look forward to receiving your revised manuscript.

Kind regards,

Muhammad Shahzad Aslam, Ph.D.,M.Phil., Pharm-D

Academic Editor

PLOS ONE

Journal Requirements:

Reviewers' comments:

Reviewer's Responses to Questions

**Comments to the Author**

1. Does the manuscript provide a valid rationale for the proposed study, with clearly identified and justified research questions?

Reviewer #2: Yes

Reviewer #3: Yes

2. Is the protocol technically sound and planned in a manner that will lead to a meaningful outcome and allow testing the stated hypotheses?

Reviewer #2: Yes

Reviewer #3: Yes

3. Is the methodology feasible and described in sufficient detail to allow the work to be replicable?

Reviewer #2: Yes

Reviewer #3: No

4. Have the authors described where all data underlying the findings will be made available when the study is complete?

Reviewer #2: Yes

Reviewer #3: Yes

5. Is the manuscript presented in an intelligible fashion and written in standard English?

Reviewer #2: Yes

Reviewer #3: Yes

6. Review Comments to the Author

You may also provide optional suggestions and comments to authors that they might find helpful in planning their study.

Reviewer #2: The justification for the exclusion criteria is clearly outlined, specifying the focus on English and Dutch studies and the rationale for excluding pandemic-specific research. The methodological rigor is strengthened by integrating the scoping review methodology with Walker and Avant’s concept analysis, ensuring a systematic and in-depth examination of the topic. To mitigate bias, the authors have included steps to address both interpretation and selection biases, enhancing the reliability of the findings. Additionally, the discussion highlights the practical relevance of the research, explaining how the results can inform policy and practice effectively. Finally, the authors mention their plans to further validate the findings within a broader research project in Flanders, underscoring the commitment to thorough and credible research outcomes.

Thank you for addressing all the concerns. Congratulations!

Reviewer #3: Please provide description of concept analysis and then the scoping review. Present their relationship with each other.

7. PLOS authors have the option to publish the peer review history of their article (what does this mean? ). If published, this will include your full peer review and any attached files.

**Do you want your identity to be public for this peer review?** For information about this choice, including consent withdrawal, please see our Privacy Policy .

Reviewer #2: **Yes: ** Jordan H. Llego, PhD ELM, D. Hon. Ex., PhDN, RN, LPT

Reviewer #3: No

---

## [Author Response · Author response to Decision Letter 2]

22 Apr 2025

Reviewer #2:

The justification for the exclusion criteria is clearly outlined, specifying the focus on English and Dutch studies and the rationale for excluding pandemic-specific research. The methodological rigor is strengthened by integrating the scoping review methodology with Walker and Avant’s concept analysis, ensuring a systematic and in-depth examination of the topic. To mitigate bias, the authors have included steps to address both interpretation and selection biases, enhancing the reliability of the findings. Additionally, the discussion highlights the practical relevance of the research, explaining how the results can inform policy and practice effectively. Finally, the authors mention their plans to further validate the findings within a broader research project in Flanders, underscoring the commitment to thorough and credible research outcomes.

Thank you for addressing all the concerns. Congratulations!

Dear reviewer, thank you for your feedback. We appreciate your positive evaluation and are glad that our adjustments are well received.

Reviewer #3:

Please provide description of concept analysis and then the scoping review. Present their relationship with each other.

Dear reviewer, thank you for your suggestion. Based on your feedback, we revised the Design section to first describe the concept analysis methodology, followed by the scoping review methodology, instead of the other way around. For each methodology, we added information about its purpose and limitations, to help clarify our rationale for combining them. This rationale for combining the methodologies is now also more explicitly stated in the final part of the section. (See also page 5, lines 92-108 of manuscript):

‘This review will use the methodological framework described by Lam Wai Shun and colleagues, which combines elements of both the concept analysis and the scoping review methodology [15]. These two methodologies will serve complementary purposes in our review.

The concept analysis methodology of Walker and Avant is a structured approach designed to explore and clarify concepts that are inconsistently used or lack conceptual clarity [14]. It offers structured guidelines for concept clarification through multiple steps: defining uses of the concept, defining attributes, antecedents, consequences, and empirical referents [14]. Nonetheless, this methodology does not specify a strategy for identifying relevant literature.

The scoping review methodology of Levac and colleagues [18] building upon the framework proposed by Arksey and O’Malley [19] provides a structured, transparent, and reproducible approach to systematically map the existing literature on a topic [18]. However, this methodology lacks specific guidelines for analyzing data for concept clarification.

By integrating these two methodologies, we can leverage the strengths of each individual methodology. The scoping review methodology ensures a systematic approach to identify and select relevant publications for our review, while the concept analysis methodology offers a structured framework for clarifying the concept of a ‘positive living environment’ in nursing homes [15]. Together they align with our goal of comprehensively exploring what constitutes a positive living environment in nursing homes.’

---

## [Decision Letter · Decision Letter 2]

24 Apr 2025

Clarifying the meaning of a positive living environment in nursing homes: a scoping review protocol

PONE-D-24-34371R2

Dear Dr. Deschodt,

We’re pleased to inform you that your manuscript has been judged scientifically suitable for publication and will be formally accepted for publication once it meets all outstanding technical requirements.

Kind regards,

Muhammad Shahzad Aslam, Ph.D.,M.Phil., Pharm-D

Academic Editor

PLOS ONE

Additional Editor Comments (optional):

Reviewers' comments:

Reviewer's Responses to Questions

**Comments to the Author**

1. Does the manuscript provide a valid rationale for the proposed study, with clearly identified and justified research questions?

Reviewer #3: Yes

2. Is the protocol technically sound and planned in a manner that will lead to a meaningful outcome and allow testing the stated hypotheses?

Reviewer #3: Yes

3. Is the methodology feasible and described in sufficient detail to allow the work to be replicable?

Reviewer #3: Yes

4. Have the authors described where all data underlying the findings will be made available when the study is complete?

Reviewer #3: Yes

5. Is the manuscript presented in an intelligible fashion and written in standard English?

Reviewer #3: Yes

6. Review Comments to the Author

You may also provide optional suggestions and comments to authors that they might find helpful in planning their study.

Reviewer #3: I admire the diligence and perseverance of the authors in doing and revising this manuscript. I hope that this will be helpful.

7. PLOS authors have the option to publish the peer review history of their article (what does this mean? ). If published, this will include your full peer review and any attached files.

**Do you want your identity to be public for this peer review?** For information about this choice, including consent withdrawal, please see our Privacy Policy .

Reviewer #3: No

---

## [Editor Report · Acceptance letter]

PONE-D-24-34371R2

PLOS ONE

Dear Dr. Deschodt,

I'm pleased to inform you that your manuscript has been deemed suitable for publication in PLOS ONE. Congratulations! Your manuscript is now being handed over to our production team.

Kind regards,

on behalf of

Dr. Muhammad Shahzad Aslam

Academic Editor

PLOS ONE